# Managing energy infrastructure to decarbonize industrial parks in China

Yang Guo [1], Jinping Tian [1]✉ & Lyujun Chen[1]✉

Industrial parks are flourishing globally and are mostly equipped with a shareable energy infrastructure, which has a long service lifetime and thus locks in greenhouse gas (GHG) emissions. We conducted a two-phase study to decarbonize Chinese industrial parks by targeting energy infrastructure. Firstly, a high-resolution geodatabase of energy infrastructure in 1604 industrial parks was established. These energy infrastructures largely featured heavy coal dependence, small capacities, cogeneration of heat and power, and were young in age. Cumulative GHG emissions, during their remaining lifetime, will reach 46.2 Gt $CO_2$ equivalent(eq.); comparable to the 11% of the 1.5 °C global carbon budget. Secondly, a vintage stock model was developed by tailoring countermeasures for each unit and implementing a cost-benefit analysis and life cycle assessment. Total GHG mitigation potential was quantified as 8%~16% relative to the baseline scenario with positive economic benefits. The synergistic reductions in freshwater consumption, $SO_2$ emissions, and $NO_x$ emissions will stand at rates of 34~39%, 24%~31% and 10%~14%, respectively.

[1] School of Environment, Tsinghua University, Beijing 100084, China. ✉email: tianjp@tsinghua.edu.cn; chenlj@tsinghua.edu.cn

Industrial parks are a common feature across countries worldwide, clustering intensive industrial activities in a tract of land[1]. Global attentions on industrial parks and their sustainability transfers are increasing in recent years[2–4]. As the world's factory, China has more than 2500 national- and provincial-level industrial parks[5], which are significant components of the industrial sector and contribute more than 50% of national industrial output[6]. Industrial park development in China was launched along with the Reform and Open Door policy in 1979 and has been central to industrialization ever since[7]. China is the largest greenhouse gas (GHG) emitter globally and promised to reach $CO_2$ emission peak around 2030[8]. Decarbonizing Chinese industrial parks will be of strong support to achieving global sustainability targets.

GHG emissions from nations[9–12], regions[13–15], and cities[16–19] have been investigated widely. Additionally, GHG accounting and mitigation in some Chinese industrial parks have been examined in the literature, such as the parks in Beijing[20], Suzhou[21,22], and Shenyang[23]. In previous work, we accounted the energy-related GHG emissions from 213 Chinese national-level industrial parks, and then uncovered GHG mitigation potentials by improving energy consumption at the park level[24]. Another study developed an assessment model to quantify GHG mitigation potentials and cost-benefits by targeting energy infrastructure in 106 Chinese eco-industrial parks[25]. However, the prospective contributions of numerous Chinese industrial parks for addressing climate change still remains unclear.

Meanwhile, energy consumption is responsible for ~60% of global GHG emissions[26]. Shareable energy infrastructure is universally used in industrial parks and generally has a long service lifetime[27–29]; thus, the GHG emissions from industrial parks are locked in. Efficient, resilient, and sustainable infrastructure is a crucial pathway to greening industrilization[30]. The energy infrastructure in an industrial park is defined as shareable utilities that are located within the park and provide energy for the park, e.g., heat and electricity[31]. Climate change mitigation requires decoupling energy services and GHG emissions. For most industrial parks, applicable GHG mitigation measures can be implemented from the perspective of energy infrastructure. Previous studies indicated that energy infrastructure accounted for an average of 75% of direct GHG emissions from industrial parks[31]. On the basis of a 30-year service lifetime[32], the in-use stocks of energy infrastructure (also called stocks[33], referring to the energy facilities in service, such as combined heat and power, heat plants, and electricity generation facilities) in Chinese industrial parks will accumulate continuously and turnover slowly. Reducing the carbon intensity of energy infrastructure in industrial parks is critical for near- and long-term climate goals.

In existing studies, GHG mitigation of industrial parks and energy infrastructure have been mostly analyzed separately, and very few studies emphasized energy infrastructure decarbonization at the industrial park level[31]. Nevertheless, these two aspects could be integrated to identify more practical and applicable options, since industrial parks have a common practice to deploy energy infrastructure, especially in China. Thus, modeling the in-use stocks of energy infrastructure in industrial parks will support assessing the effectiveness of GHG mitigation options in a practical way, because reasonable GHG mitigation measures should be cost-effective and involve less environmental burden shifting. Thus, to provide a comprehensive and delicate decision-making basis, the environmental impacts associated with GHG mitigation actions need to be determined to identify the synergy or tradeoffs among these factors, such as water consumption and air pollutant emissions. In addition, there is still a lack of bottom-up databases regarding Chinese industrial parks to support such study.

To provide the full spectrum of GHG mitigation in Chinese industrial parks by managing energy infrastructure, first, this study uncovered the energy infrastructure stocks of 1604 industrial parks in China. We established a high-resolution geodatabase of 4706 energy infrastructure units in these parks, including their geographic coordinates, vintages (the year of energy infrastructure starting working), capacities, fuel consumption, technologies, efficiencies, etc. This inventory covered all the authorized national- and provincial-level industrial parks on the official catalog issued in 2006[34]. Second, an integrated assessment model, called the vintage-stock model, was developed from the insight of the stocks' vintages, which integrated measure-unit matchmaking, scenario setup, energy efficiency assessment, GHG emission accounting, cost-benefit analysis, and life-cycle assessment. The model quantified the GHG mitigation potentials, economic costs, material consumption (concrete, steel, iron, and aluminum), and environmental co-benefits (water saving, $SO_2$ emission reductions, and $NO_x$ emission reductions) of decarbonizing the energy infrastructure stocks in the parks. In doing so, environmental burden shifting and indirect environmental impacts can be explored for a systematic analysis.

## Results

**Energy infrastructure in Chinese industrial parks.** The geodatabase of energy infrastructure in 1604 Chinese industrial parks covered 2127 plants, including 4706 units. Fig. 1 illustrates the overview of energy infrastructure in the parks by the end of 2014, from the perspective of stock evolution, fuel structure, and capacity structure. The geographic locations of the parks and facilities are presented in Supplementary Fig. 1. Among these parks, 850 were equipped with shareable energy infrastructure, consisting 4542 in-use units. The total capacity of the stocks was 515 GW, accounting for 38% of China's installed electricity generation capacity in 2014. Considering there is no hydropower utility in these parks, the ratio of total energy infrastructure capacity to non-hydropower installed capacity in China was nearly half (48%). We identified that 87% of the stocks in terms of capacity were coal-fired units, indicating that the parks are more coal-dependent than the national average level (61% of the national installed electricity capacity was coal-fired in 2014[35]). Then natural gas-fired stocks were second to coal, contributing 8.2% of the total capacity. The renewable energy-driven units had a capacity ratio of 0.92%, including biomass, biogas, wind energy, solar energy, and geothermal energy. The remaining stocks, 4.1%, were based on other fuels, such as diesel, coal gangue, coal gas, waste heat, municipal solid waste (MSW), and sludge.

As for vintage years, 58% of the stocks in capacity started working from 2001 to 2010. Driven by national energy policies[36], the new units established since 2007 mostly have a large capacity, 300 MW or more. Large utilities with a capacity of 300 MW or more accounted for 72% of the total capacity but only 19% of the total number of units. In contrast, small units with a capacity no greater than 30 MW accounted for 7% of the total capacity but 59% of the total number. Generally, large-capacity units outperform small ones in terms of energy efficiency[37]; thus, there is significant GHG mitigation potential by sizing-up small utilities. Regarding energy outputs, combined heat and power (CHP) was the main form of the stocks, accounting for 48% of the total capacity and 61% of the total number of units.

Furthermore, there was a significant difference among the parks in terms of the total installed capacity of energy infrastructure, ranging from 1.2–6706 MW. Most of the units located in coastal areas (see Supplementary Fig. 1), indicated by that Liaoning, Hebei, Shandong, Jiangsu, Zhejiang, and Guangdong Provinces host 47% of the total capacity. The coal-fired energy infrastructure

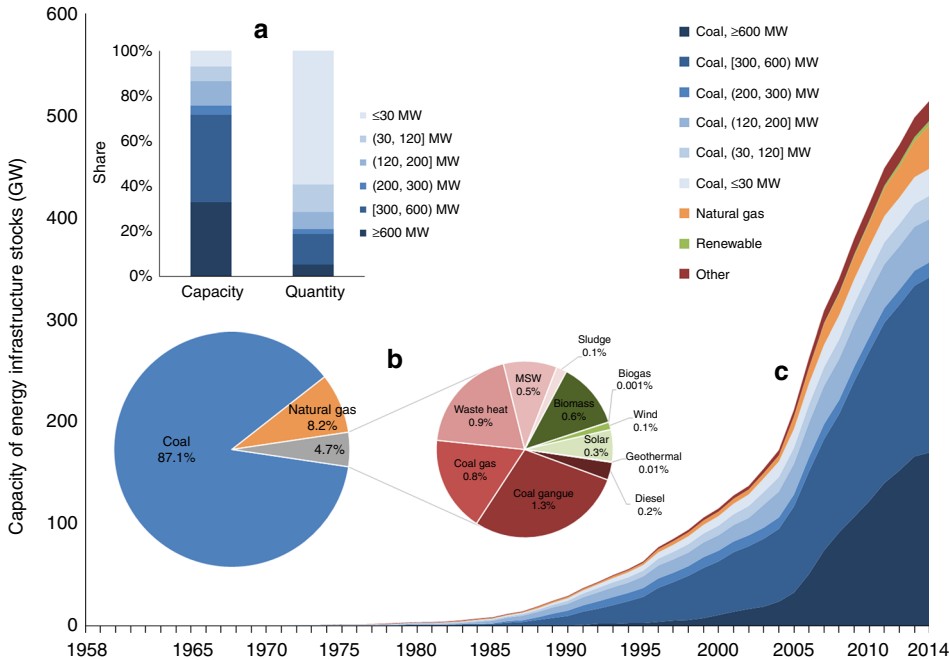

**Fig. 1 In-use stocks of energy infrastructure in the 1604 Chinese industrial parks. a** The inner bar chart refers to the organization of stocks in terms of the rated capacity and number of units. **b** The pie chart refers to the fuel structure of stocks. **c** The stack curve refers to the cumulative capacity of the stocks, which was calculated as the stock in last year + the added capacity in this year—the retired capacity in this year. Source data are provided as a Source Data file.

in the parks increased rapidly from 2004–2010 and then slowed down after 2011. Coal-fired power utilities in China were over-installed[38], which led to a continuous decrease in average annual working time[39]. Therefore, the construction of new coal-fired utilities has been strictly supervised and controlled since the 13th 5-year Plan Period (2016–2020)[40,41], resulting in the above trajectory of stock accumulation for coal-fired units in these industrial parks.

**Comprehensive environmental impacts**. By applying the environmental impact accounting method proposed in this study, the annual GHG emissions, freshwater consumption, $SO_2$ emissions, and $NO_x$ emissions of energy infrastructure stocks in the 1604 Chinese industrial parks were quantified as 2.35 Gt $CO_2$ equivalant (eq.)., 3.43 $Gm^3$, 2.35 Mt, and 3.07 Mt, accounting for 18% of national GHG emissions, 4.6% of national industrial water consumption, 12% of national $SO_2$ emissions, and 15% of national $NO_x$ emissions, respectively[42–44]. These results demonstrate the critical role of industrial parks in addressing environmental issues in China. The cumulative GHG emissions from the stocks during their remaining lifetime was 46.2 Gt $CO_2$ eq., equivalent to 11% of the 1.5 °C global carbon budget (a remaining budget of about 420 Gt $CO_2$ for a two-thirds chance of limiting warming to 1.5 °C)[45]. In addition, the average remaining lifetime of the stocks covers 2015–2032, which is temporally consistent with the 2030 carbon peak target of China. Thus, decarbonizing these stocks will contribute substantially and continuously to achieving the China's Intended Nationally Determined Contributions.

Energy infrastructure is the key node of metabolism systems in industrial parks, linking energy, water, and pollutants together[46,47]. The environmental impacts of energy infrastructure can be a reasonable proxy of the whole park to a great extent[31]. The GHG emissions, freshwater consumption, $SO_2$ emissions, and $NO_x$ emissions of the stocks in the parks are shown in Fig. 2 and Supplementary Figs. 2–4, respectively, with

geographical details. Most GHG emissions were generated from the highly developed industrial parks located in coastal areas, such as Liaoning, Hebei, Shandong, Jiangsu, Zhejiang, and Guangdong Provinces. These six provinces were responsible for 51% of the total GHG emissions from the stocks. From the fuel perspective, coal-fired units emitted 92.3% of the total GHG emissions, while natural gas-fired, coal gas-fired, coal gangue-fired, MSW-driven, and other units emitted 2.8%, 2.2%, 1.9%, 0.5%, and 0.4% of the total GHG emissions, respectively. From the capacity perspective, units with a capacity of >= 600 MW, [300, 600) MW, and <300 MW accounted for 31%, 33%, and 36% of the total GHG emissions, respectively. In particular, coal-fired units with a capacity <300 MW were responsible for 30% of the total GHG emissions, 37% of the total freshwater consumption, 49% of the total $SO_2$ emissions, and 35% of the total $NO_x$ emissions, indicating that these units should be regarded as prior candidates for improving environmental performance.

**GHG mitigation potentials and cost-effectiveness**. By examining the features of the in-use energy infrastructure stocks, best-practice measures for GHG mitigation in the parks[31], and national energy strategies[48–50], we proposed five key GHG mitigation measures by considering the feasibility and applicability for most parks: M1, retrofitting coal-fired boilers to natural gas (NG)-fired boilers; M2, replacing coal-fired boilers with MSW incinerators; M3, retrofitting extraction-condensing or pure-condensing turbines to back-pressure turbines; M4, replacing small-capacity coal-fired units with large-capacity coal-fired units; and M5, replacing small-capacity coal-fired units with large-capacity natural gas combined cycle (NGCC) units. Each unit is assessed individually to confirm the appropriate measure by considering technical attributes (fuel, technology, capacity, efficiency, and vintage) and additional requirements (such as fuel consumption quota and geographic proximity). Fig. 3 summarizes the measure-unit matchmaking briefly. The detailed

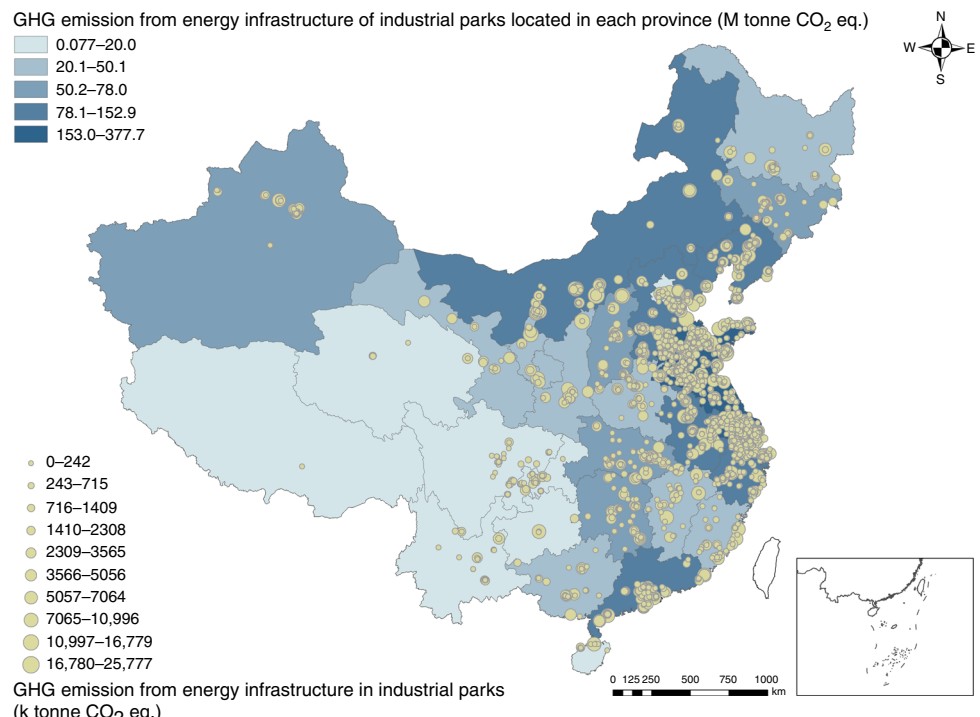

**Fig. 2 GHG emissions from energy infrastructure in the 1604 Chinese industrial parks.** The beige circle symbol size is proportional to the greenhouse gas (GHG) emissions from energy infrastructure in industrial parks, and the color depth is positively correlated with the GHG emissions from energy infrastructure of industrial parks in each provincial area. Source data are provided as a Source Data file.

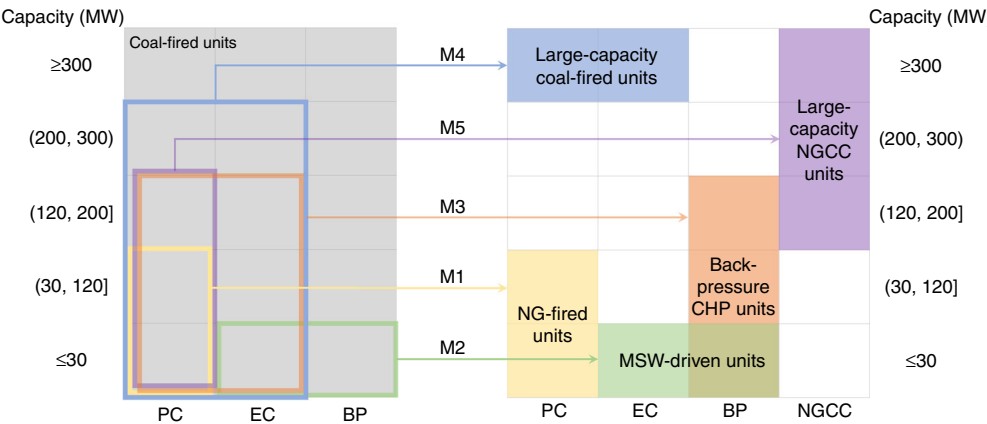

**Fig. 3 Matchmaking between GHG mitigation measures and energy infrastructure units.** PC, pure condensing; EC, extraction condensing; BP, back pressure; NGCC, natural gas combined cycle; CHP, combined heat and power; and MSW, municipal solid waste. M1–M5 work on their appropriate candidates in different capacities and technologies (framed on the left), by retrofitting or replacing them into the units shown on the right.

descriptions for M1–M5 and their clients are provided in the Supplementary Notes 1-7.

A baseline scenario and six mitigation scenarios were established to quantify the GHG mitigation potentials, costs, and environmental co-benefits of decarbonizing the energy infrastructure stocks by using the vintage-stock model. In the baseline scenario, the in-use stocks will keep their original attributes and continue working until they are retired normally. In the mitigation scenarios, the measures M1–M5 will be implemented separately (single-measure scenario) or jointly (integrated scenario) to modify their appropriate clients. The GHG mitigation potentials, environmental co-benefits (freshwater saving, $SO_2$ reductions, and $NO_x$ reductions), economic costs, and material consumption in mitigation scenarios are presented in Table 1. The environmental benefits included both direct (on-site emissions or consumption) and

indirect portions (off-site/upstream impacts of the production and transportation processes for energy, water, and materials). More details are elaborated in the Methods section and Supplementary Notes 1-7.

The results indicated that all the single-measure scenarios had positive GHG mitigation potentials (315~2304 $MtCO_2$ eq.) and negative costs (−1041~−96 CNY/$tCO_2$ eq.). M3 had the largest GHG potential among M1–M5 due to noticeable efficiency improvement and extensive client units. M2 was accompanied by the most significant GHG mitigation benefit, which comes from MSW treatment subsidies, coal cost savings, and a high feed-in tariff for MSW-based electricity. Besides, almost all the GHG mitigation measures will bring the environmental co-benefits of reducing freshwater consumption, $SO_2$ emissions, and $NO_x$ emissions. The material consumption levels of M4 or M5 were

**Table 1 Environmental benefits, economic costs, and material consumption in GHG mitigation scenarios.**

| Indicator | | M1 | M2 | M3 | M4 | M5 | Integrated |
|---|---|---|---|---|---|---|---|
| Environmental benefit | $\Delta$GHG (MtCO$_2$ eq.) | −315 | −1254 | −2304 | −1200 | −1207 | −4688 |
| | $\Delta$Freshwater (Mm$^3$) | 178 | 1631 | −20,233 | −8472 | −2508 | −24,804 |
| | $\Delta$SO$_2$ (kt) | −1133 | −2950 | −3805 | −8299 | −3023 | −11,742 |
| | $\Delta$NO$_x$ (kt) | −948 | 483 | −3426 | −4728 | −1875 | −6152 |
| | $\Delta$GHG (%) | −0.68 | −2.71 | −4.99 | −2.60 | −2.61 | −10.15 |
| | $\Delta$Freshwater (%) | 0.27 | 2.49 | −30.94 | −12.95 | −3.84 | −37.93 |
| | $\Delta$SO$_2$ (%) | −2.52 | −6.55 | −8.45 | −18.44 | −6.72 | −26.09 |
| | $\Delta$NO$_x$ (%) | −1.64 | 0.84 | −5.92 | −8.17 | −3.24 | −10.63 |
| Economic cost (B CNY) | $\Delta$Equipment cost | 6.53 | 59.6 | 7.17 | 229 | 44.7 | 197 |
| | $\Delta$Fuel cost | 156 | −1016 | −1015 | −431 | 191 | −1445 |
| | $\Delta$Electricity benefit | 191 | 176 | | | 669 | 954 |
| | Total net cost | −29.0 | −1132 | −1007 | −202 | −434 | −2202 |
| | Unit Cost (CNY/t) | −96 | −1,041 | −479 | −182 | −382 | −516 |
| Material consumption (k t) | $\Delta$Concrete | | | | 12,035 | 1660 | 6326 |
| | $\Delta$Steel | 2.79 | 119 | 496 | 3845 | 527 | 2489 |
| | $\Delta$Iron | | | | 46.9 | 6.93 | 25.1 |
| | $\Delta$Aluminum | | | | 31.8 | 3.46 | 15.8 |

$\Delta$ = variation compared with the baseline scenario; % refers to the variation rates in the GHG mitigation scenarios compared with the baseline scenario; the costs and benefits are accounted at the constant price of 2015.

much greater than those of M1-M3 because of establishing new utilities rather than retrofitting existing utilities. If these five measures were implemented together (the integrated scenario), the reductions of GHG, freshwater, SO$_2$ and NO$_x$ will be significant compared with those of the baseline scenario, with reduction rates of 10.1%, 37.9%, 26.1%, and 10.6%, respectively. The detailed indicators of each scenario are computed and listed in the Source Data. Uncertainty analysis clarified that the reduction rate ranges in the integrated scenario compared with the baseline scenario were 8–16%, 34–39%, 24–31%, and 10–14% for GHG, freshwater, SO$_2$ and NO$_x$, respectively. By integrating the five measures to decarbonize the energy infrastructure in the 1604 Chinese industrial parks, the cumulative GHG mitigation potential during their remaining lifetime in the integrated scenario was 4.69 Gt CO$_2$ eq., equivalent to 1.1% of the 1.5 °C global carbon budget[45].

Further, the environmental impacts associated with GHG mitigation measures were decomposed by unit capacity, as shown in Fig. 4. The client units with a capacity of ≤ 30 MW, (30, 60] MW, and (120, 200] MW contributed the majority in almost all scenarios, e.g., 91% of the total GHG mitigation potentials, 81% of the total freshwater savings, 89% of the total SO$_2$ reductions, and 86% of the total NO$_x$ reductions in the integrated scenario. M3 had the most substantial GHG mitigation potential due to the efficiency improvements from extraction-condensing or pure-condensing technologies to back-pressure technology. The freshwater consumption in M1 and M2, and the NO$_x$ emissions in M2 were increased compared with those in the baseline scenario. This is led by the freshwater consumption factors of NG and MSW combustion and the NO$_x$ emission factor of MSW combustion were higher than those of coal combustion.

Moreover, the environmental benefits of the GHG mitigation scenarios were divided into direct and indirect portions as Table 2. The direct portion referred to on-site improvements through energy saving and alternatives, unit efficiency enhancement and technology upgrades, while the indirect portion referred to upstream environmental impacts avoided or increased by energy saving, water saving, and material consumption. In all the GHG mitigation scenarios, the majority of environmental benefits were direct, but the indirect portion cannot be overlooked. For instance, in the integrated scenario, the indirect portions accounted for 9%, 6%, 5%, and 15% of the total amount of mitigated GHG emissions, freshwater savings, SO$_2$ reductions and NO$_x$ reductions, respectively. Thus, incorporating the indirect portions by a life-cycle thinking contributes to identifying the environmental impacts of GHG mitigation measures systematically.

## Discussion

Peaking CO$_2$ emissions in some industrial parks before 2030 has been highlighted as a vital target in the green development strategies issued by the Chinese government[51]. Thus, for industrial parks, much efforts should be made to decouple their economic development and GHG emissions. This study, along with previous works[25,31], has definitely suggested that the energy infrastructure in China's industrial parks can play a pivotal role in mitigating GHG emissions with alleviating air pollution and saving water simultaneously. Reducing coal dependence and improving efficiencies of energy infrastructure in industrial parks are critical pathways toward sustainability targets. We mapped the comprehensive environmental impacts and projected improvements of energy infrastructure stocks based on a second to none high-resolution inventory and a delicate assessment model. Specifically, these in-use stocks of energy infrastructure were featured by more small-capacity coal-fired CHP units than those outside industrial parks. The GHG emissions, water consumption, and air pollutant emissions from the energy infrastructure stocks are uncovered as remarkable shares of the whole country.

Accordingly, infrastructure-tailored countermeasures for GHG mitigation were designed and planned in a robust and applicable way for industrial parks. These actions were on the basis of an accurate unit-measure matchmaking and could be expected to deliver a real-world impact on China's industrial parks. Further, the vintage-stock model combined the stocks' remaining lifetime and life-cycle assessment to provide the decarbonization results in term of various indicators for policy consideration. In most cases, the GHG mitigation measures are accompanied with a negative economic cost and positive impacts on freshwater saving and air pollutant emissions. These findings could support the feasibility of such countermeasures to promoting climate change mitigation and green industrial development in China. This work will also be meaningful globally, particularly in other developing countries

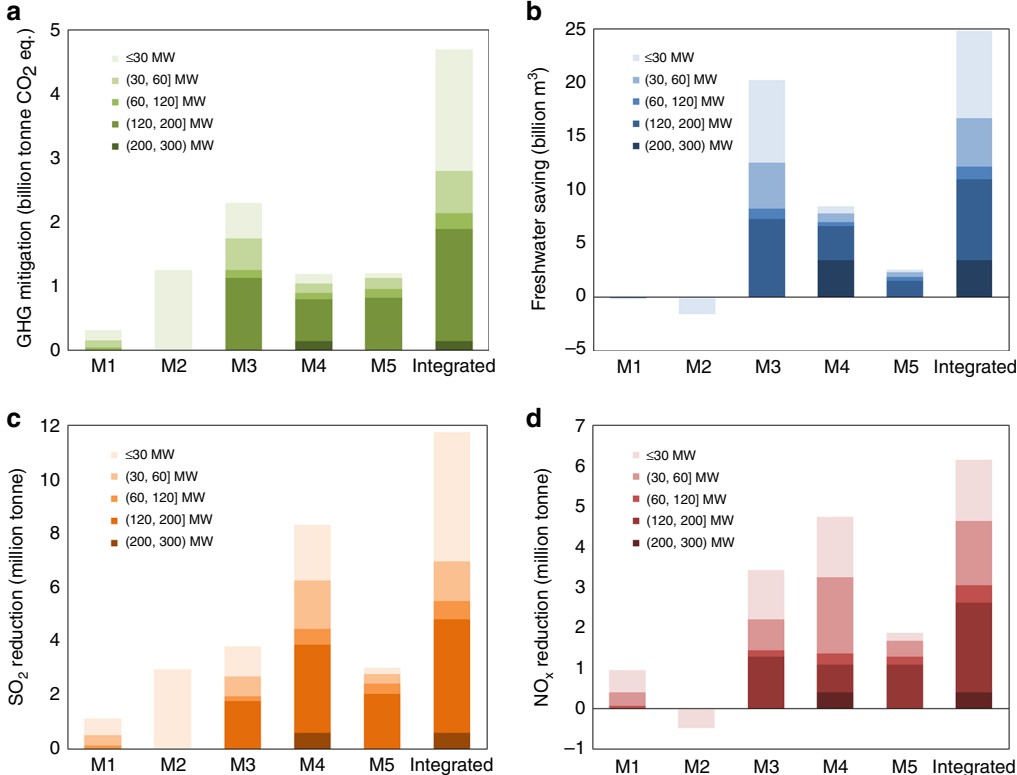

**Fig. 4 Environmental benefits of GHG mitigation scenarios decomposed by different-sized units. a–d** present GHG mitigation potential, freshwater saving, SO2 emission reduction, and NOx emission reduction in M1–M5 and Integrated scenarios. M1, retrofitting coal-fired boilers to natural gas (NG)-fired boilers; M2, replacing coal-fired boilers with MSW incinerators; M3, retrofitting extraction-condensing or pure-condensing turbines to back-pressure turbines; M4, replacing small-capacity coal-fired units with large-capacity coal-fired units; and M5, replacing small-capacity coal-fired units with large-capacity natural gas combined cycle (NGCC) units; Integrated scenario, implementing M1–M5 jointly. Source data are provided as a Source Data file.

**Table 2 Direct and indirect environmental benefits in GHG mitigation scenarios.**

| Indicator | | M1 | M2 | M3 | M4 | M5 | Integrated |
|---|---|---|---|---|---|---|---|
| Direct | $\Delta$GHG (MtCO$_2$ eq.) | −302 | −1087 | −2101 | −1110 | −1137 | −4266 |
| | $\Delta$Freshwater (Mm$^3$) | 159 | 1917 | −19,086 | −8184 | −2429 | −23,284 |
| | $\Delta$SO$_2$ (kt) | −1141 | −2730 | −3476 | −8187 | −2992 | −11,196 |
| | $\Delta$NO$_x$ (kt) | −890 | 820 | −3010 | −4546 | −1662 | −5209 |
| Indirect | $\Delta$GHG (MtCO$_2$ eq.) | −13.6 | −167 | −202 | −90.1 | −70.1 | −422 |
| | $\Delta$Freshwater (Mm$^3$) | 19.0 | −286 | −1,148 | −287 | −79.2 | −1,521 |
| | $\Delta$SO$_2$ (kt) | 8.42 | −220 | −329 | −112 | −31.1 | −547 |
| | $\Delta$NO$_x$ (kt) | −58.5 | −337 | −416 | −182 | −213 | −943 |

that are facilitating low-carbon industrial parks to improve industrialization and urbanization.

Currently, this study conducted the decarbonization of industrial parks by optimizing energy infrastructure itself, not embedding other sectors into a comprehensive analysis. However, incorporating other infrastructures in industrial parks, such as wastewater treatment plants, to establish a symbiotic infrastructure system will contribute to deep decarbonization actions and environmentally beneficial energy-water nexus. Besides, symbiotic linkages between infrastructure and other energy-intensive industrial sectors, e.g., steel and cement industries, can be advantageous to explore cross-sector sustainability measures, and industrial parks will be suitable sites to implement these connections.

## Methods
**Geodatabase of energy infrastructure in Chinese industrial parks.** The database of energy infrastructure stocks in 1604 Chinese industrial parks was

established and embedded into a geographic information system (GIS) framework. Three steps were used to identify the stocks located in these parks by 2014. First, the lists of industrial parks[34] and in-use energy facilities[35] in China were mapped separately using ArcGIS software. The geographical coordinates of the parks and facilities were gathered from Google Earth. In China, each industrial park has an administrative organization (called an administrative committee) that is in charge of park management. The coordinate of the administrative committee of an industrial park was used as the proxy of the park. Second, a circle with a radius of R was drawn at the center of the park administrative committee. Then, the energy facilities standing within the circle were screened out. The R value was set as 10 km, which is the appropriate service distance recommended by national CHP guidelines[52]. By doing so for all the parks, a preliminary inventory of the energy facilities in the industrial parks was established. Third, the preliminary inventory of energy facilities derived in the second step was validated one by one to confirm whether a certain energy facility was in a park. The approaches for this confirmation included on-site investigations and questionnaires for more than 200 parks, a literature review, and information extraction from official reports and websites of the parks, energy facility project websites, and environmental impact assessment reports of CHP projects. This step was the most time-consuming component.

Based on the inventory established above, additional data such as capacities, vintage years, energy efficiencies, technologies, fuel inputs, and energy outputs of the energy infrastructure stocks, were mined out from multiple sources. The data

sources mainly included the following: statistical data of Chinese electricity industry;[35] manuals of power-generating units in China;[53] basic information tables for thermal power-generating units at capacity levels of 100–225 MW, 300 MW and 600 MW;[54–56] lists of desulfurization and denitrification facilities of coal-fired power-generating units;[57] on-site investigations and questionnaires for more than 200 industrial parks; more than 100 eco-industrial park reports and park websites; and more than 500 environmental impact assessment reports of CHP facilities issued by local governments. Then, a bottom-up and high-resolution geodatabase of 4706 energy infrastructure units in 1604 Chinese industrial parks was built.

**Integrated vintage-stock model**. The vintage-stock model integrates measure-unit matchmaking, scenario setup, energy efficiency assessment, GHG emission accounting, cost-benefit analysis, and life-cycle assessment. The model can quantify the GHG mitigation potentials, costs, and environmental co-benefits of decarbonizing energy infrastructure. The vintage-stock model is an updated version of our previous model[25], by improving the following aspects. First, indirect environmental impacts derived from a life-cycle perspective are incorporated for a more systematic assessment; second, environmental co-benefits (freshwater saving, $SO_2$ emission reduction, and $NO_x$ emission reduction) and material consumption (concrete, steel, iron, and aluminum) are quantified accompanied with GHG mitigation potentials and costs; third, the matchmaking criteria between GHG mitigation measures and client units are improved in a more practical way, by considering the proper capacities of units for implementing each measure, natural gas consumption quota, planned added capacity for municipal solid waste incineration, etc. Fig. 5 clarifies the model framework, which consists of three modules, scenario setup, impact assessment, and multi-indicator accounting. Seven scenarios were proposed, including the baseline, single-measure (M1–M5), and integrated scenarios.

M1–M5 were matched with energy infrastructure units according to their attributes, such as capacity, vintage, technology, fuel, efficiency, and location. All units were set to have a 30-year service lifetime based on national guidelines[32]. In the baseline scenario, all in-use stocks will retire normally. In the GHG mitigation measures (i.e., M1–M5 and integrated scenarios), an overnight retrofitting for the client units is assumed at $T^0$ before the stocks retire. $T^0$ represents the time when each measure begins for the appropriate clients, which was 2015 in this study. Thus, the effective period for M1, M2, and M3 was $[T^0, v_{ij} + 30]$, and that for M4 and M5 was $[T^0, T^0 + 30]$, where $v_{ij}$ is the vintage year (such as 2001) of the jth unit in the ith plant. All the GHG mitigation measures are implemented simultaneously in the integrated scenario.

The GHG mitigation costs involved four components as Δequipment costs, Δfuel costs, Δelectricity benefits, and Δheat benefits. The Δ symbol refers to the differences between the GHG mitigation scenarios and the baseline scenario. Additionally, water consumption, air pollutant emissions, and material consumption also vary in association with the GHG mitigation measures due to changes in capacity, technology, fuel type, end-of-pipe treatment, etc. Such a carbon-water-emission nexus should be considered for a more systematic assessment. Thus, both the direct and indirect environmental impacts were accounted from a life-cycle perspective. Specifically, in addition to the on-site variations in GHG emissions, freshwater consumption, $SO_2$ emissions, and $NO_x$ emissions, the off-site environmental impacts embodied in the upstream production and transportation processes of fuel, water, and materials were included. Detailed descriptions of the model can be found in the Supplementary Notes 1–7 with Supplementary Figs. 5–7 and Supplementary Tables 5–21.

**Uncertainty analysis**. The uncertainty in the model results was mainly sourced from the small proportion of units lacking detailed operation parameters, such as annual working time, energy efficiency, and heat-to-electricity ratio. These data-missing units were categorized by capacity, fuel, and technology; then, the default values of the corresponding indicators of data-available units in our database were used for the data-missing units. Thus, uncertainty analysis was conducted by using a Monte Carlo simulation for the key results, i.e., GHG mitigation rates, freshwater saving rates, $SO_2$ reduction rates, and $NO_x$ reduction rates in the M1–M5 and integrated scenarios. The operation parameters of the data-missing units varied randomly and simultaneously in the uniform distribution of [minimum, maximum]. The statistical values of each parameter for each category of units were listed in Supplementary Tables 1–3. The average or median values were chosen to be the default to calculate the original results.

By further analyzing the related equations in the model, the key results monotonously increased or decreased with the operation parameters, including boiler efficiencies, heat-to-electric ratios, effective electrical efficiencies, annual working hours, and self-use electricity rates. Thus, the minimum and maximum values of the key results were derived by applying two-end values of the operation parameters (i.e., the minimum and maximum values of data-available units). The reduction rates for GHG, water, $SO_2$, and $NO_x$ changed simultaneously with parameter variation, and their two-end values were not achieved at the same time. That is, when the GHG mitigation rates reached the maximum values, the water saving rates may reach the minimum values. In this study, GHG mitigation was the primary goal; thus, the uncertainty analysis was centered on GHG mitigation rates. The two-end values of reduction rates in GHG, water, $SO_2$, and $NO_x$ are listed in

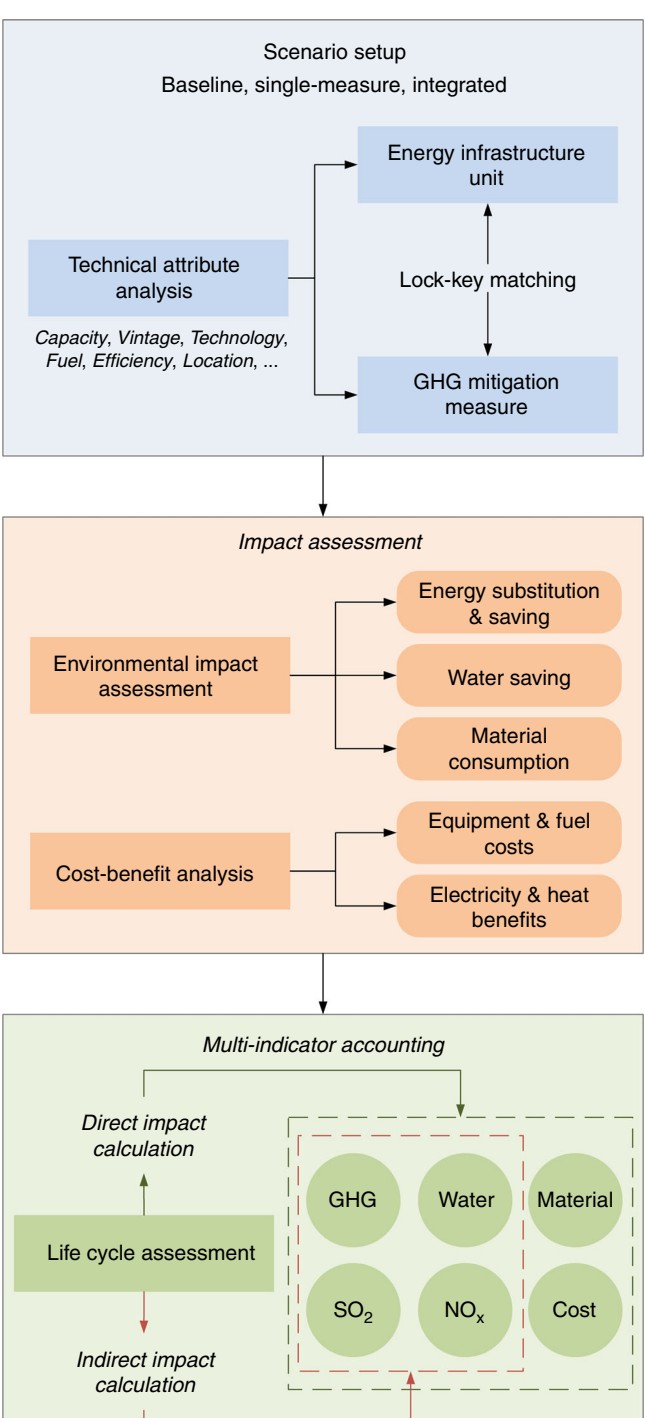

**Fig. 5 Framework of vintage-stock model.** Measure-unit matchmaking, scenario setup, energy efficiency assessment, GHG emission accounting, cost-benefit analysis, and life cycle assessment are integrated in the model to quantify the GHG mitigation potentials, economic costs, and environmental co-benefits of decarbonizing energy infrastructures during their remaining lifetimes.

Supplementary Table 4. In the integrated scenario, the ranges in reduction rates were 8–16%, 34–39%, 24–31%, and 10–14% for GHG emissions, freshwater consumption, $SO_2$ emissions, and $NO_x$ emissions, respectively.

**Reporting summary**. Further information on research design is available in the Nature Research Reporting Summary linked to this article.

## Data availability

The source data underlying Figs. 1, 2, 4 and Supplementary Figs. 1–4 are provided as a Source Data file. All data used for this analysis are available from publicly available sources cited or from the authors upon reasonable request. Publicly accessible data sources are basic information tables for thermal power-generating units at capacity levels of 100–225, 300, and 600 MW (http://www.chinapower.com.cn/kjfwduibiao/20160422/4824.html and http://kjfw.cec.org.cn/kejifuwu/2013-04-07/99877.html), and lists of desulfurization and denitrification facilities of coal-fired power-generating units (http://www.mee.gov.cn/gkml/hbb/bgg/201407/t20140711_278584.htm).

## Code availability

The vintage-stock model we developed is connected to source data. Computer code is available from the authors upon request.

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

## Acknowledgements

The authors acknowledge the National Natural Science Foundation in China for its financial support through projects 41671530, 41971267, and 41471468. Yang Guo acknowledges the financial support of PLC 2018–2019 Thesis Scholarship from Peking University-Lincoln Institute Center for Urban Development and Land Policy. The authors are grateful to Ms. Na Zang, Yang Gao, Wanying Lu, Xiaoxue Tian, and Mr. Xing Li for collecting data from industrial parks.

## Author contributions

L.C., J.T., and Y.G. designed the study; Y.G., J.T., and L.C. compiled the database; Y.G., J.T., and L.C. developed the vintage-stock model; Y.G. conducted the calculations and drew the figures; Y.G. led the writing, and all the authors revised the manuscript together.

## Competing interests

The authors declare no competing interests.
