## [Peer Review File · Nature Communications]

Reviewers' comments:

Reviewer #1 (Remarks to the Author):

This paper established a high-resolution geodatabase of energy infrastructure in Chinese industrial parks and used a vintage stock model to uncover the GHG mitigation potentials. The topic is interesting and important. I would like to recommend its publication if the following issues are solved well.

Major concerns:

1. This paper is similar to the authors' previous work (Guo et al., 2016) on both the propose and the model. Authors need to demonstrate clearly the academic contributions of this work.
2. There are too many exogenous parameters, which result into large uncertainties. The uncertainty analysis needs to be more discussed and moved to the main text.

Specific comments:

1. The Abstract of this paper was too long. Abstract is the epitome of the whole paper, which reflects the theme, purpose and the main/core findings of the paper.
2. In the introduction section, the authors didn't clearly describe the necessity and contributions of the research. Although the authors reviewed some studies, the contributions of the paper were unclear and confused.
3. Moreover, it is desirable to add a conclusion part to summarize the research from both theoretical and practical significance.
4. Some terms in this paper should to be defined. The meaning of these terms that can be understood only by those who are already familiar with them. The reader who does not know them would need more details. For example:
 - What does the "energy infrastructure stocks" mean? Is the "stock" a kind of unit?
 - What does the "vintages" represent for? and the description of "vintage-stock mode" in Line 270-271 need to be adjusted to the place where the term first appeared in the text.
5. In Figure 2, in my view, (b), (c) and (d) could move to the Supplementary Information.
6. The English language needs to be polished. Several sentences of this paper are unclear or grammatically incorrect, which need to be modified.

References:

Guo, Y., Tian, J., Chertow, M., Chen, L. Greenhouse gas mitigation in Chinese eco-industrial parks by targeting energy infrastructure: A vintage stock model. *Environmental Science & Technology*, 50, 11403-11413 (2016).

Reviewer #2 (Remarks to the Author):

The study is interesting and rrelevant in the current context. However, a more generic problem as well theoretical arguments about the adopted approach are necessary for avoiding the simple case study situation.

Reviewer #3 (Remarks to the Author):

The paper presents a detailed evaluation of the GHG emissions mitigation potential of decarbonizing energy infrastructure within 1064 Chinese industrial parks. A total of 4542 energy units are included in the study, accounting for 38% of the installed electricity generation capacity in China. The units are mainly coal fueled, of small capacity, involving cogeneration of heat and power, and recently installed.

Five decarbonizing measures are considered both separately and jointly applied, involving

retrofitting or replacing of the existing energy infrastructures. A cumulative GHG mitigation potential, including reduction in freshwater consumption, and in SO₂ and NO_x emissions, of about 10% respect to the baseline scenario (emissions of the units during their remaining service lifetime) results from the study.

The research proceeds through successive steps. First, a geo-referenced inventory of the energy infrastructures installed in the industrial parks is compiled, mapping and matching the two items. The environmental impact of the energy infrastructures, as they are, is evaluated to depict the baseline scenario. Then, the assessment model (called vintage-stock model) was improved and applied to evaluate the impact of the selected decarbonizing measures. The model integrates the matchmaking of the decarbonizing measure to the infrastructure unit, the scenario setup, the energy efficiency assessment, GHG emission accounting, cost-benefit analysis and life cycle assessment.

General remarks:

The key result of the paper is the demonstration of a viable and effective strategy to reduce the carbon footprint of the industrial parks in China. Due to the fast and massive industrial growth of the last decades, China is one of the major GHG emissions contributor and the Chinese government is now strongly committed to support measures aimed at mitigating environmental impacts of the large-scale industrial districts typical of the country 1, 2. Thus, the relevance of the result rests on providing a possible pathway for decarbonizing the industrial parks at systemic level. Moreover, the industry sector is strongly required to reduce the GHG emissions both in advanced and emerging economies, and the sustainability transition schemes of industrial parks have been widely studied 3. This makes the paper of interest to researchers, industry managers and decision makers involved in energy management and low carbon options for the industrial parks.

The result is supported by a convincing methodology, including uncertainty analysis, and a very plentiful and valuable inventory, clearly described in section 5 (Data and Methods) and extensively detailed in the supplementary material, where a complete dataset and the exhaustive model description are provided.

Specific comments:

- The abstract is clear, but too long (more than 300 words) respect to the Journal submission instructions (about 150 words), and should be shorten to improve its sharpness. As an example, the data listed in lines 14 to 16 could be moved to the next session.
- The introduction clarifies the context and the scope of the paper. However, some weak points should be improved:
 - o In line 34 the reference [4] is very old and the following more recent ones, showing the global attention on the industrial parks and their sustainable transformation, should be added:
 - Towards sustainable business parks: A literature review and a systemic model
 - Co-benefits accounting for the implementation of eco-industrial development strategies in the scale of industrial park based on emergy analysis
 - A socio-ecological approach to improve industrial zones towards eco-industrial parks
 - o Similarly, in line 38 the following more recent references should be added:
 - Renewable energy in eco-industrial parks and urban-industrial symbiosis: A literature review and a conceptual synthesis
 - Symbiosis between industrial systems, utilities and public service facilities for boosting energy and resource efficiency
 - o The references in line 48 refer only to China. What about other nations, regions and cities?
 - o Lines 57-58: please add at least one reference supporting your statement.
- The Results and Discussion sections are not clearly emphasized: a clearer breakdown and headings, according to the Journal scheme, would improve the results presentation. Moreover, while the results are extensively presented (classification of mapped energy infrastructures, present and potential environmental impacts), the discussion and conclusions section is rather poor (lines 228-239). The discussion section should include the outline of the key results underlining their relevance and applicability, the main limitations of the study and the conclusions.

1. Huang, B. et al. Review of the development of China's Eco-industrial Park standard system. *Resour. Conserv. Recycl.* 140, 137–144 (2019).
2. Yu, X., Chen, H., Wang, B., Wang, R. & Shan, Y. Driving forces of CO₂ emissions and mitigation strategies of China's National low carbon pilot industrial parks. *Appl. Energy* 212, 1553–1562 (2018).
3. Susur, E., Hidalgo, A. & Chiaroni, D. A strategic niche management perspective on transitions to eco-industrial park development: A systematic review of case studies. *Resour. Conserv. Recycl.* 140, 338–359 (2019).

Point-to-point response to the comments

Response to Reviewer #1

Major Concern 1:

This paper established a high-resolution geodatabase of energy infrastructure in Chinese industrial parks and used a vintage stock model to uncover the GHG mitigation potentials. The topic is interesting and important. I would like to recommend its publication if the following issues are solved well.

Response:

We appreciate your invaluable suggestions on the manuscript. The manuscript is improved extensively by considering your comments.

Major Concern 2:

This paper is similar to the authors' previous work (Guo et al., 2016) on both the propose and the model. Authors need to demonstrate clearly the academic contributions of this work.

Response:

Thanks for the comment. Compared with our previous work, this paper is an extensively new study from the perspective of intensive database enrichment, new model development, and substantial improvement of. Each aspect is clarified in detail as follows.

First, our understanding on the roles and countermeasures of mitigating GHGs in China's industrial parks is a step-by-step process, from limited samples to almost all parks considered. In this new work, we established a high-resolution geodatabase covering 1604 industrial parks in China and containing all the parks in the Catalog of Chinese industrial parks (2006). Our previous work sampled 106 eco-industrial parks, which have relatively better environmental performance among all the parks. Thus,

the newly extended database can represent the overall profile of China's industrial parks and lay a solid basis to yield data-driven policy implications targeting low-carbon development for both industrial parks and industrial sectors in China. Moreover, the inventories of data also have been enriched besides GHG-related data, the newly added data include the technologies of cooling, desulfurization, and denitrification, freshwater consumption for cooling and heat supply, SO₂ emission, and NO_x emission.

Second, we developed a new version of vintage-stock model which has been substantially updated as compared with the previous model, mainly in the following aspects: (1) Indirect environmental impacts embodied in the upstream production and transportation processes of fuel, water, and materials are considered from a life-cycle perspective; (2) Freshwater consumption, SO₂ emissions, NO_x emissions, and material consumption (concrete, steel, iron, and aluminum) are included; and (3) The matchmaking criteria between GHG mitigation measures and client units are updated and improved by considering more practical factors, including the proper unit capacities for implementing each measure, unit vintage, natural gas consumption quota, and planned added capacity for municipal solid waste incineration. Thus, the novelty and robustness of the model can be substantially advanced in a more systematic and practical way.

Third, we clearly quantified the comprehensive environmental impacts (GHG emission, freshwater consumption, SO₂ emissions, and NO_x emission) of energy infrastructure in 1604 parks and their contributions to the whole country. We also explored the extra impacts associated with decarbonizing the energy infrastructure, including the environmental co-benefits (freshwater saving, SO₂ emission reduction, and NO_x emission reduction) and material consumption (concrete, steel, iron, and aluminum) accompanied with GHG mitigation potentials and costs. Such environmental co-benefits were further decomposed to direct and indirect portions from a life-cycle perspective and by different capacity-level units. By doing so, the policy visions and practical implications were informed in a much clear manner, by identifying the most appropriate candidate units to be retrofitted or replaced in

priority.

The advances of this study mentioned above and the comparison with our previous work is summarized in the following table. Accordingly, we enriched the related clarifications in the text, especially in Section 4 (Data and Methods).

	Database	Methods	Results
Previous work	548 units in 106 Chinese eco-industrial parks	Integrating unit-measure matchmaking, efficiency assessment, GHG emission accounting, cost benefit analysis, and scenario setup	GHG mitigation potential and costs
This study	4706 units in 1604 Chinese industrial parks, covering all the parks listed in the Catalog of Chinese Industrial Parks (2006)	Besides integrating all the modules in previous work, the new model is updated by follows: (1) embedding life cycle assessment, environmental co-benefits accounting, and material consumption accounting into the model; and (2) updating matchmaking criteria by incorporating the proper unit capacity levels for implementing each measure, unit vintage, natural gas consumption quota, and newly planned capacity for municipal solid waste incineration.	(1) Diverse environmental impacts are considered simultaneously, including GHG emission, freshwater consumption, SO ₂ emission, and NO _x emission; (2) besides GHG mitigation potentials and costs, a wide range of results are explored, including environmental co-benefits (freshwater saving, and the reductions of SO ₂ emission and NO _x emission), and material consumption (concrete, steel, iron, and aluminum); and (3) Direct and indirect portions of GHG mitigation potentials and environmental co-benefits are further presented from a life-cycle perspective.

Specific Comment 3:

There are too many exogenous parameters, which result into large uncertainties. The uncertainty analysis needs to be more discussed and moved to the main text.

Response:

Your comment is much appreciated. We improved the uncertainty analysis by enriching the references, sources and considerations for all the exogenous parameters of our model, and considering both technical parameters and policy targets. These additions are properly placed in both the text and Supplementary Information. Besides, we moved the uncertainty analysis to the main text as Section 4.3 (Uncertainty analysis).

Specific Comment 4:

The Abstract of this paper was too long. Abstract is the epitome of the whole paper, which reflects the theme, purpose and the main/core findings of the paper.

Response:

We have revised the abstract to make it concise and sharp, and present the theme, purpose and core findings clearly and directly. The length has been shortened to about 190 words.

Specific Comment 5:

In the introduction section, the authors didn't clearly describe the necessity and contributions of the research. Although the authors reviewed some studies, the contributions of the paper were unclear and confused.

Response:

Thanks for the comment. The introduction section has been restructured and enriched to highlight the necessity, novelty and contributions of this study. In short, the study fill the following gaps:

- 1) China is the largest GHG emitter, and decarbonizing Chinese industrial parks will be of strong support to achieving global sustainability targets. However, the prospective contributions of Chinese industrial parks for addressing climate change still remains unclear, due to significant number and diversity of the parks. Thus, a high-resolution and intensive-sample based study is very necessary.
- 2) Energy infrastructure is widely employed in industrial parks and is the main source of GHG emission from industrial parks. In existing studies, GHG mitigation of industrial parks and energy infrastructure have been mostly analyzed separately. Nevertheless, these two aspects could be integrated to identify more practical and applicable options, since industrial parks are primary sites to deploy energy infrastructure, especially in China.

Specific Comment 6:

Moreover, it is desirable to add a conclusion part to summarize the research from both theoretical and practical significance.

Response:

We have embedded the conclusions into the Discussion (Section 3) to summarize the theoretical and practical significance of this work, due to no separate conclusion section is requested by the journal instructions.

Specific Comment 7:

Some terms in this paper should to be defined. The meaning of these terms that can be understood only by those who are already familiar with them. The reader who does not know them would need more details. For example:

- *What does the “energy infrastructure stocks” mean? Is the “stock” a kind of unit?*
- *What does the “vintages” represent for? and the description of “vintage-stock*

mode” in Lines 270-271 need to be adjusted to the place where the term first appeared in the text.

Response:

- 1) ‘Energy infrastructure stocks’ is the short expression for the in-use stocks of energy infrastructure, which refers to the energy infrastructure in service within the park, such as combined heat and power, steam plants, electricity generation facilities. We added this explanation and related references to the first place where this term appears (lines 45-46 in the clean version of revised manuscript).
- 2) ‘Vintage’ refers to the year of energy infrastructure starting working. We supplemented this statement in the first place where this term appears (line 64 of the clean text). Besides, we moved the description for vintage stock model to the first place of term appearing (lines 67-68 of the clean text).

Specific Comment 8:

In Figure 2, in my view, (b), (c) and (d) could move to the Supplementary Information.

Response:

We moved these three sub-figures to the Supplementary Information as Figures S6-S8.

Specific Comment 9:

The English language needs to be polished. Several sentences of this paper are unclear or grammatically incorrect, which need to be modified.

Response:

Thanks for the comment. We have improved the language of the manuscript through a professional English editing, the ACS ChemWorx team.

Response to Reviewer #2

Remark:

The study is interesting and relevant in the current context. However, a more generic problem as well as theoretical arguments about the adopted approach are necessary for avoiding the simple case study situation.

Response:

We appreciate your approval of our work as well as the insightful comment. We have clearly highlighted the general research question proposed and the method adopted as follows, which have been also embedded into the Introduction (Section 1).

This study aims to investigate the general strategies of decarbonizing industrial parks, by targeting the energy infrastructure that has been widely employed in the parks and contributed a major part of GHG emission from the parks. Industrial parks are a common feature of industrial development across countries worldwide by clustering intensive industrial activities in a tract of land. China is the biggest GHG emitter and has intensive industrial parks. Decarbonizing Chinese industrial parks will be of strong support to achieving global climate change targets. Therefore, we performed this study by zooming in the Chinese industrial parks as demonstration for other countries and then delivered global implications.

As for methodology, in existing studies, GHG mitigation of industrial parks and energy infrastructure have been mostly studied separately. Nevertheless, these two aspects could be integrated to identify more practical and applicable countermeasures, since industrial parks are primary sites to deploy energy infrastructure. Modeling the in-use stocks of energy infrastructure (also called “stocks”, referring to the energy facilities in service, such as combined heat and power, steam plants, and electricity generation facilities) in industrial parks will support assessing the effectiveness of GHG mitigation options in a reliable way. Thus, we developed an integrated assessment model from the insight of the stocks’ service lifetime, called the “vintage stock model”, which integrated measure-unit matchmaking, scenario setup, energy

efficiency assessment, GHG emission accounting, cost-benefit analysis, and life cycle assessment. The model quantified not only GHG mitigation potentials and costs of decarbonizing the energy infrastructure in industrial parks, but also material consumption (concrete, steel, iron, and aluminum) and environmental co-benefits (water saving, SO₂ reduction, and NO_x reduction). In doing so, the environmental burden shifting and indirect environmental impacts can be incorporated for a systematic analysis.

Response to Reviewer #3

General Remark:

The paper presents a detailed evaluation of the GHG emissions mitigation potential of decarbonizing energy infrastructure within 1064 Chinese industrial parks. A total of 4542 energy units are included in the study, accounting for 38% of the installed electricity generation capacity in China. The units are mainly coal fueled, of small capacity, involving cogeneration of heat and power, and recently installed.

Five decarbonizing measures are considered both separately and jointly applied, involving retrofitting or replacing of the existing energy infrastructures. A cumulative GHG mitigation potential, including reduction in freshwater consumption, and in SO₂ and NO_x emissions, of about 10% respect to the baseline scenario (emissions of the units during their remaining service lifetime) results from the study.

The research proceeds through successive steps. First, a geo-referenced inventory of the energy infrastructures installed in the industrial parks is compiled, mapping and matching the two items. The environmental impact of the energy infrastructures, as they are, is evaluated to depict the baseline scenario. Then, the assessment model (called vintage-stock model) was improved and applied to evaluate the impact of the selected decarbonizing measures. The model integrates the matchmaking of the decarbonizing measure to the infrastructure unit, the scenario setup, the energy efficiency assessment, GHG emission accounting, cost-benefit analysis and life cycle

assessment.

The key result of the paper is the demonstration of a viable and effective strategy to reduce the carbon footprint of the industrial parks in China. Due to the fast and massive industrial growth of the last decades, China is one of the major GHG emissions contributor and the Chinese government is now strongly committed to support measures aimed at mitigating environmental impacts of the large-scale industrial districts typical of the country^{1,2}. Thus, the relevance of the result rests on providing a possible pathway for decarbonizing the industrial parks at systemic level. Moreover, the industry sector is strongly required to reduce the GHG emissions both in advanced and emerging economies, and the sustainability transition schemes of industrial parks have been widely studied³. This makes the paper of interest to researchers, industry managers and decision makers involved in energy management and low carbon options for the industrial parks.

The result is supported by a convincing methodology, including uncertainty analysis, and a very plentiful and valuable inventory, clearly described in section 5 (Data and Methods) and extensively detailed in the supplementary material, where a complete dataset and the exhaustive model description are provided.

Response:

Thank you for the careful review and in-depth understanding of our study. We also appreciate your highlighting the significance and approval for both the database and methodology. We carefully studied your comments, and improved the manuscript extensively as follows.

Specific Comment 1:

The abstract is clear, but too long (more than 300 words) respect to the Journal submission instructions (about 150 words), and should be shorten to improve its sharpness. As an example, the data listed in lines 14 to 16 could be moved to the next session.

Response:

We have shortened the abstract to about 190 words for a better sharpness, by embedding less significant sentences (such as those in lines 14-16) to the Introduction and other sections.

Specific Comment 2:

The introduction clarifies the context and the scope of the paper. However, some weak points should be improved:

- *In line 34 the reference [4] is very old and the following more recent ones, showing the global attention on the industrial parks and their sustainable transformation, should be added:*

- Towards sustainable business parks: A literature review and a systemic model*

- Co-benefits accounting for the implementation of eco-industrial development strategies in the scale of industrial park based on emergy analysis*

- A socio-ecological approach to improve industrial zones towards eco-industrial parks*

- *Similarly, in line 38 the following more recent references should be added:*

- *Renewable energy in eco-industrial parks and urban-industrial symbiosis: A literature review and a conceptual synthesis*

- *Symbiosis between industrial systems, utilities and public service facilities for boosting energy and resource efficiency*

- *The references in line 48 refer only to China. What about other nations, regions and cities?*

- *Lines 57-58: please add at least one reference supporting your statement.*

Response:

Thanks for these suggestions. The five recommended references has been added in the appropriate places, in lines 21 and 38 of the clean text. We have also supplemented five references for GHG emissions at the national, regional, and city levels besides China, as well as the reference supporting the statement in line 58 of original text (line 51 in the clean version of revised text).

Specific Comment 3:

The Results and Discussion sections are not clearly emphasized: a clearer breakdown and headings, according to the journal scheme, would improve the results presentation. Moreover, while the results are extensively presented (classification of mapped energy infrastructures, present and potential environmental impacts), the discussion and conclusions section is rather poor (lines 228-239). The discussion section should include the outline of the key results underlining their relevance and applicability, the main limitations of the study and the conclusions.

Response:

We have adjusted the headings of results and discussion to comply with the journal guidelines. The text has been revised to the structure as follows.

1. *Introduction*
2. *Results*
 - 2.1 *In-use stocks of energy infrastructure in Chinese industrial parks*
 - 2.2 *Comprehensive environmental impacts of energy infrastructure stocks*
 - 2.3 *GHG mitigation potentials, costs, and environmental co-benefits of energy infrastructure stocks*
3. *Discussion*
4. *Data and methods*
 - 4.1 *Geodatabase of energy infrastructure in Chinese industrial parks*
 - 4.2 *Integrated vintage stock model*
 - 4.3 *Uncertainty analysis*

Moreover, we have enriched the Discussion (Section 3) accordingly, as:

“Peaking CO₂ emission in some industrial parks before 2030 has been listed as a significant target of the green development strategies issued by the Chinese government⁵¹. Thus, for industrial parks, much efforts should be made to decouple their economic development and GHG emissions. This study, along with previous works^{25,31}, has definitely suggested that the energy infrastructure in China’s industrial parks can play a pivotal role in mitigating GHG emissions with alleviating air pollution and saving water simultaneously. Reducing coal dependence and improving efficiencies of energy infrastructure in industrial parks are critical pathways toward

sustainability targets. We mapped the comprehensive environmental impacts and projected improvements of energy infrastructure stocks based on a second to none high-resolution inventory and a delicate assessment model. Specifically, these in-use stocks of energy infrastructure were featured by more small-capacity coal-fired CHP units than those outside industrial parks. The GHG emission, water consumption, and air pollutant emissions from the energy infrastructure stocks are uncovered as remarkable shares of the whole country.

Accordingly, infrastructure-tailored countermeasures for GHG mitigation were designed and planned in a robust and applicable way for industrial parks. These actions were on the basis of an accurate unit-measure matchmaking and could be expected to deliver a real-world impact on China's industrial parks. Further, the vintage stock model combined the stocks' remaining lifetime and life cycle assessment to provide the decarbonization results in term of various indicators for policy consideration. In most cases, the GHG mitigation measures are accompanied with a negative economic cost and positive impacts on freshwater saving and air pollutant emissions. These findings could support the feasibility of such countermeasures to promoting climate change mitigation and green industrial development in China. This work will also be meaningful globally, particularly in other developing countries that are targeting low-carbon industrial parks to improve industrialization and urbanization.

Currently, this study conducted the decarbonization of industrial parks by optimizing energy infrastructure itself, not embedding other sectors into a comprehensive analysis. However, incorporating other infrastructures in industrial parks, such as wastewater treatment plants, to establish a symbiotic infrastructure system will contribute to deep decarbonization actions and environmentally beneficial energy-water nexus. Besides, symbiotic linkages between infrastructure and other energy-intensive industrial sectors, e.g., steel and cement industries, can be advantageous to explore cross-sector sustainability measures, and industrial parks will be suitable sites to implement these connections.”

REVIEWERS' COMMENTS:

Reviewer #1 (Remarks to the Author):

My comments have been solved well. The manuscript can be considered for publication.

Reviewer #3 (Remarks to the Author):

The revised paper is improved accordingly to the presented specific comments.

- The shortened abstract results more incisive,
- the introduction is better contextualized and the references have been updated and enriched
- the headings adjusting makes the manuscript purposes clearer
- the improved discussion section underlines the importance of the research in relation to the current global discussion on the pathway to reduce the environmental impact of industry and industrial park

The quality of the revised paper makes the manuscript suitable for publication.

Point-to-point response to the comments - R2

Response to Reviewer #1

Remark:

My comments have been solved well. The manuscript can be considered for publication.

Response:

We appreciate your invaluable suggestions to help us improve the quality of our manuscript. Your approval of our work is also much thankful.

Response to Reviewer #3

Remark:

The revised paper is improved accordingly to the presented specific comments.

- *The shortened abstract results more incisive,*
- *the introduction is better contextualized and the references have been updated and enriched*
- *the headings adjusting makes the manuscript purposes clearer*
- *the improved discussion section underlines the importance of the research in relation to the current global discussion on the pathway to reduce the environmental impact of industry and industrial park*

The quality of the revised paper makes the manuscript suitable for publication.

Response:

Thank you very much for your considerate comments. Through revising our manuscript carefully according to your suggestions, the manuscript has been re-conducted to comply with the instructions and quality for publication, and the novelty and contribution of our work have been better highlighted in the context of industrial sustainability and climate change mitigation.